# Edit3D: Elevating 3D Scene Editing with Attention-Driven Multi-Turn Interactivity

## ABSTRACT

With the rise of new 3D representations like NeRF and 3D Gaussian splatting, creating realistic 3D scenes is easier than ever before. However, the incompatibility of these 3D representations with existing editing software has also introduced unprecedented challenges to 3D editing tasks. Although recent advances in text-to-image generative models have made some progress in 3D editing, these methods either lack precision or require users to manually specify the editing areas in 3D space, complicating the editing process. To overcome these issues, we propose Edit3D, an innovative 3D editing method designed to enhance editing quality. Specifically, we propose a multi-turn editing framework and introduce an attention-driven open-set segmentation (ADSS) technique within this framework. ADSS allows for more precise segmentation of parts, which enhances the editing precision and minimizes interference with pixels in areas that are not being edited. Additionally, we propose a fine-tuning phase, intended to further improve the overall editing quality without compromising the training efficiency. Experiments demonstrate that Edit3D effectively adjusts 3D scenes based on textual instructions. Through continuous and multiple turns of editing, it achieves more intricate combinations, enhancing the diversity of 3D editing effects.

## CCS CONCEPTS

• **Computing methodologies → 3D imaging**.

## KEYWORDS

3D editing, attention-driven, multi-turn editing

## 1 INTRODUCTION

Traditional 3D editing is a complex task, typically requiring professional software to precisely adjust models' shape and texture. This process demands expertise from the user and makes the entire editing procedure complicated. Recently, innovative 3D representations such as NeRF [36] and 3D Gaussian splatting [20] have provided unprecedented support for the reconstruction of 3D scenes. Although these emerging 3D representations have facilitated the reconstruction of real 3D scenes, they also present new challenges: their incompatibility with existing 3D editing software means that traditional editing methods are no longer applicable. As generative artificial intelligence progresses, there is an need to develop new

*ACM MM, 2024, Melbourne, Australia*

© 2024 Copyright held by the owner/author(s). Publication rights licensed to ACM.
ACM ISBN 978-x-xxxx-xxxx-x/YY/MM
https://doi.org/10.1145/nnnnnnn.nnnnnnn

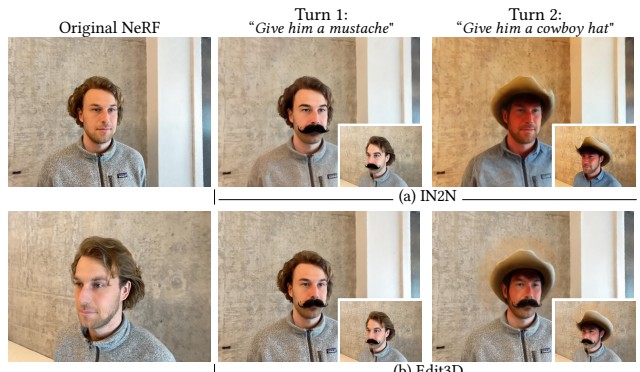

**Figure 1: Two-step 3D editing comparison. The IN2N method adds a mustache but later removes it when adding a hat, also blurring the clothing and background. The Edit3D method adds both items without affecting clarity, showing its effectiveness in sequential edits.**

tools and methods to edit and adjust the latest 3D representations both easier and more intuitive.

Text-to-image generative models offer new technological means for 3D editing techniques [14, 30, 34, 43]. Among these innovative methods, IN2N [14] has garnered widespread attention for its text-driven 3D editing capabilities. This technology allows for the editing of 3D scenes through text instructions, greatly simplifying the 3D editing process. Despite the notable advancements for IN2N in 3D editing, its editing capabilities are based on the InstructPix2Pix model [2]. The InstructPix2Pix model uses a diffusion process to edit images, processing the entire image as input, causing all pixels in the image to undergo a diffusion process. This global editing may cause unintended alterations in certain areas not intended for change, causing the final 3D scene to diverge from the editor's original intent. To further enhance the effects of 3D editing, it is necessary to optimize the current editing framework to ensure precise operations in target areas during the editing process.

In this paper, we propose a text-driven 3D editing method called Edit3D and have implemented three improvement measures. First, we improve the IN2N algorithm. The optimized algorithm can adapt to multiple turns of continuous editing processes, allowing users to achieve more complex 3D editing effects by combining a series of editing instructions. Second, we integrate a segmentation model into the 3D editing framework, which enhanced the precision of editing operations and ensured that the edits accurately target the intended locations. Existing open-set segmentation techniques do not perform well in handling fine-grained segmentation tasks. To address this, we propose an innovative attention-driven open-set segmentation (ADSS) method that achieves more refined object segmentation, thereby enabling more precise editing of 3D scenes.

Throughout the study, we observe an intriguing phenomenon: when the model is given precise image text descriptions (captions), it can generate more accurate attention maps, which in turn improves the accuracy of image segmentation tasks. This further confirms the importance and potential applications of textual information in the field of image recognition. Finally, we introduce a carefully designed fine-tuning phase to further enhance the overall effect of 3D editing, while ensuring that the extra training costs stay reasonable.

Fig. 1 shows an example of a two-turn editing process. In this example, the IN2N approach introduces some unnecessary changes to areas not targeted for editing when executing the editing instructions. For example, IN2N initially succeeds in adding a mustache to the character after the first stage. In the second turn of editing, a cowboy hat is further added. However, while new elements are added, the previously added mustache is removed. Besides, the character's clothing and the image background also become less clear. In contrast, with our Edit3D method, not only are the mustache and hat successfully added, but the clarity of the clothing and background is well maintained. This indicates that our Edit3D method not only achieves single editing effects with greater precision but also produces more complex results through multiple cascading edits.

## 2 RELATED WORK

Progress in using instructions as an editing interface has evolved in parallel with the development of large language models (LLMs) [3, 42, 44–46, 58, 64]. LLMs enable users to complete tasks via textual instructions. Based on diffusion-based generative models [7, 17, 18, 52, 54], text-to-image models facilitate the creation of images from textual prompts [12, 40, 48, 50, 66]. This technology has advanced to real images by identifying and modifying the word embeddings, followed by regenerating the image with the intended edits [10, 39, 47, 49, 53]. InstructPix2Pix [2] simplifies the editing workflow by employing Prompt-to-Prompt [16] for training dataset generation, resulting in an instruction-based model that intuitively edits images via text prompts. Extending this concept, IN2N [14] has pioneered instruction-based editing for 3D scenes [36]. Unlike the multi-turn dialogues facilitated by LLMs in neural language understanding, instruction-based editing in image and 3D domains has been confined to single-turn interactions [1, 19]. Single-turn interactions differ from their multi-turn counterparts as they operate on a single sentence or prompt without maintaining context or coherence across the conversation [59]. Our method enhances instruction-based editing in the 3D domain to encompass multi-turn scenarios, thereby achieving a consistent and coherent editing experience throughout successive editing interactions.

In instruction-based image editing, a common issue is the unintended alteration of areas outside the target region. To address this, open-set detection/segmentation models [8, 13, 23, 24, 28, 31, 37, 63] capable of accurately identifying editing areas can be utilized. However, these models, while proficient in detecting a wide range of object categories, often lack the part-level recognition ability. For more granular editing, part-level recognition is essential. Current fine-grained segmentation models [6, 29, 35, 56, 68], such as those trained on datasets like Pascal-Part [4] and PartImageNet [15], have

limitations due to dataset annotation biases. We observe that the SAM model [21] demonstrates robust segmentation performance across various datasets. However, for segmenting specific parts, SAM necessitates additional inputs like points or boxes. Our proposal involves using the SAM model for initial robust segmentation. This process can be augmented with vision-language multimodal models [22, 26, 27, 32, 60] to locate specific parts via attention maps [51, 67]. Given the capabilities shown by multimodal models like GPT-4V [41], we anticipate further improvements in the proposed open-set segmentation framework with the release of more advanced multimodal models in the research community.

## 3 METHOD

This paper focuses on the improvement of 3D editing techniques, without being specific to any particular form of 3D representation. The editing strategies proposed are universal, capable of being applied to various representations such as NeRF [36] and 3D Gaussian splatting [20]. To elaborate on the strategy specifically, this paper selects NeRF as an example of 3D data representation. In Sec. 3.1, we improve the single-turn editing strategy of the IN2N model to a multi-turn editing method. Subsequently, in Sec. 3.2, we adopt attention-driven segmentation technology to precisely confine the area of editing operations, ensuring the accuracy of the edits. Finally, in Sec. 3.3, we apply mixed super-resolution technology to enhance the editing effects, further improving the detail and visual quality of the edited 3D scene.

### 3.1 Overcoming Catastrophic Forgetting

IN2N simplifies the process of editing NeRF with text instructions, allowing for editing to be made in a single turn. A more natural method, however, is to edit 3D scenes through a series of turns. For instance, IN2N performs effectively in the initial turn of a two-stage editing process, as illustrated in Fig. 1(a), where it successfully adds a mustache to a figure following the given instruction. Yet, when a new instruction is introduced in the subsequent turn, such as "*Give him a cowboy hat*", the system only applies this latest instruction and overlooks the previous edit of adding the mustache. This phenomenon is known as catastrophic forgetting. In this section, we discuss how we improve the IN2N framework by integrating capabilities that allow for consecutive editing, thereby resolving the issue of catastrophic forgetting. Specifically, the IN2N framework edits images rendered by NeRF using InstructPix2Pix, conditioned by three key elements: a reference image $C_{\text{image}}$, a text instruction $C_{\text{text}}$, and a noise-injected input $\mathbf{z}_t$. The process is encapsulated by a function $G(\cdot)$, yielding the modified image as $G(C_{\text{image}}, C_{\text{text}}, \mathbf{z}_t)$.

**Pre-Rendered Images as Condition.** During training, the IN2N model uses the original dataset image $I_{\text{dataset}}^v$ as the conditioning image $C_{\text{image}}$, where $v$ is the camera's calibrated viewpoint. It generates a new image from viewpoint $v$ using the current NeRF model, labeled $I_{\text{rendered}}^v$. For the noisy input $\mathbf{z}_t$, since the InstructPix2Pix framework operates in a latent space, $I_{\text{rendered}}^v$ is first encoded by a VAE encoder $\mathcal{E}$ and then combined with Gaussian noise to create $\mathbf{z}_t$, where $t$ represents the degree of noise added. To simplify, we refer to this encoding and noising process as $F_t(\cdot)$, resulting in $\mathbf{z}_t = F_t(I_{\text{rendered}}^v)$. Consequently, the edited image by IN2N is given

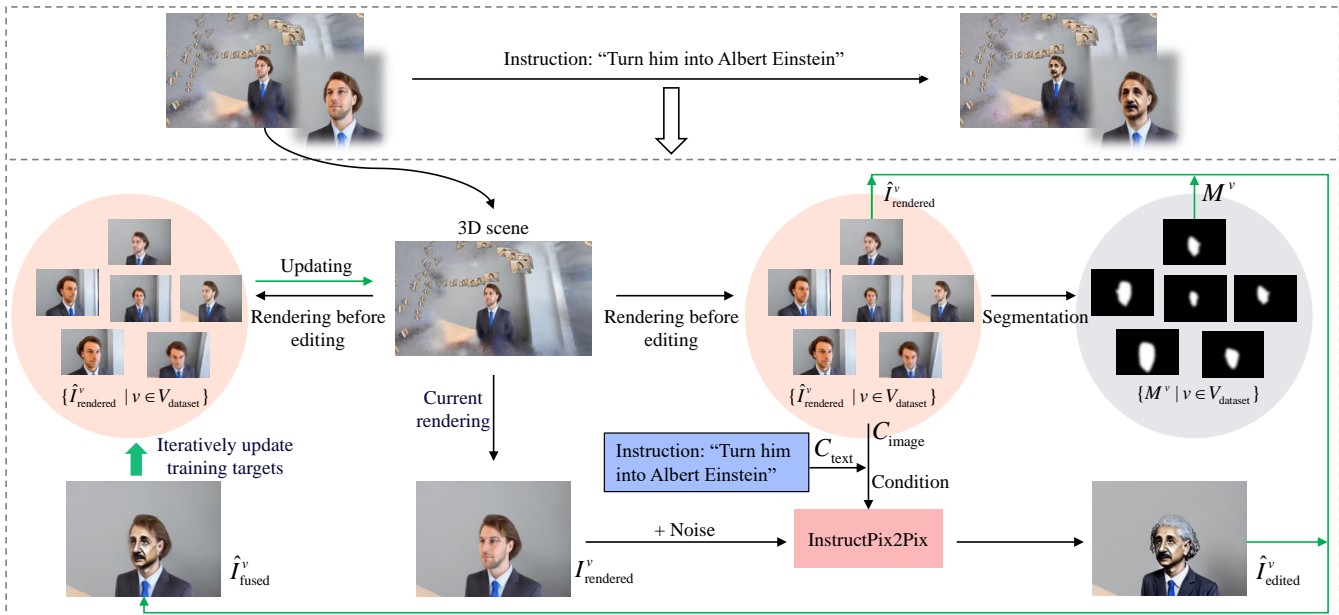

**Figure 2: Overview of the Edit3D method. Our method begins with pre-rendered images at the start of each editing session, which are used as the condition for InstructPix2Pix, and as the initial guiding images for supervising the 3D scene to prevent catastrophic forgetting (as detailed in Sec. 3.1). Additionally, semantic segmentation is applied to address the issue of characteristic drift (explained in Sec. 3.2).**

by

$$I_{\text{edited}}^v = G(I_{\text{dataset}}^v, C_{\text{text}}, F_t(I_{\text{rendered}}^v)). \tag{1}$$

According to Eq. (1), the edited image in the current turn, denoted as $I_{\text{edited}}^v$, integrates information from both the conditioning image $I_{\text{dataset}}^v$ and the textual instruction $C_{\text{text}}$. During the multi-turn editing process, changes made in earlier turns are captured by $I_{\text{rendered}}^v$, yet the addition of noise compromises the preservation of these edits, interrupting the continuity. To address this, we start each editing turn by pre-rendering a set of images $\{\hat{I}_{\text{rendered}}^v \mid v \in V_{\text{dataset}}\}$ using the updated NeRF from the preceding turn. The set $V_{\text{dataset}}$ encompasses all camera viewpoints in the dataset. It is important to note that the pre-rendered image $\hat{I}_{\text{rendered}}^v$ is distinct from $I_{\text{rendered}}^v$ mentioned in Eq. (1). The former maintains the modifications from prior turns, whereas the latter, updated by the current NeRF, might not retain the impact of those earlier changes. To overcome this, we utilize $\hat{I}_{\text{rendered}}^v$ as the conditioning image, and thus the edited image according to our method is given by:

$$\hat{I}_{\text{edited}}^v = G(\hat{I}_{\text{rendered}}^v, C_{\text{text}}, F_t(I_{\text{rendered}}^v)). \tag{2}$$

Fig. 2 illustrates the schematic of the editing framework. It is important to recognize that Eq. (1) represents a specific instance of Eq. (2) in the context of a single-turn editing scenario. Assuming the NeRF has converged on the original dataset, the rendered image $\hat{I}_{\text{rendered}}^v$ is approximately equal to $I_{\text{dataset}}^v$. As a result, the edited image $\hat{I}_{\text{edited}}^v$ closely resembles $I_{\text{edited}}^v$. In scenarios involving multiple editing turns, utilizing the previously rendered image $\hat{I}_{\text{rendered}}^v$

instead of the original dataset image $I_{\text{dataset}}^v$ as the conditioning image ensures that edits from earlier stages are seamlessly integrated into subsequent ones.

**Pre-Rendered Images as Targets.** The edited images created by InstructPix2Pix exhibit inconsistencies when viewed from different viewpoints, making them unsuitable for direct use in supervising the training of the 3D representation. To resolve this, IN2N introduces an Iterative Dataset Update (*Iterative DU*) algorithm, which incrementally incorporates edited images into the original dataset. During the training phase, NeRF randomly selects rays from a range of viewpoints. Consequently, the ground truth pixel values for these rays could be derived either from the original dataset or from the edited images. It has been demonstrated that this method of mixed supervision leads to more stable training and yields multi-view consistent results in 3D editing.

*Iterative DU* is effective for single-turn editing, but it's not suitable for scenarios involving multiple editing turns. In these cases, the current NeRF already incorporates the editing effects of previous turns. Therefore, using images from the original dataset to supervise the current NeRF could erase the edits made in earlier turns. To address this, we begin each editing session with a set of pre-rendered images $\{\hat{I}_{\text{rendered}}^v \mid v \in V_{\text{dataset}}\}$, where $\hat{I}_{\text{rendered}}^v$ represents the edits from earlier sessions. We employ these pre-rendered images as training targets for the current NeRF. At the same time, we gradually replace these targets with the newly edited images $\hat{I}_{\text{edited}}^v$, as specified in Eq. (2). This approach ensures that the NeRF during each editing phase is guided either by the previously rendered $\hat{I}_{\text{rendered}}^v$ or the recently edited images $\hat{I}_{\text{edited}}^v$. Since both

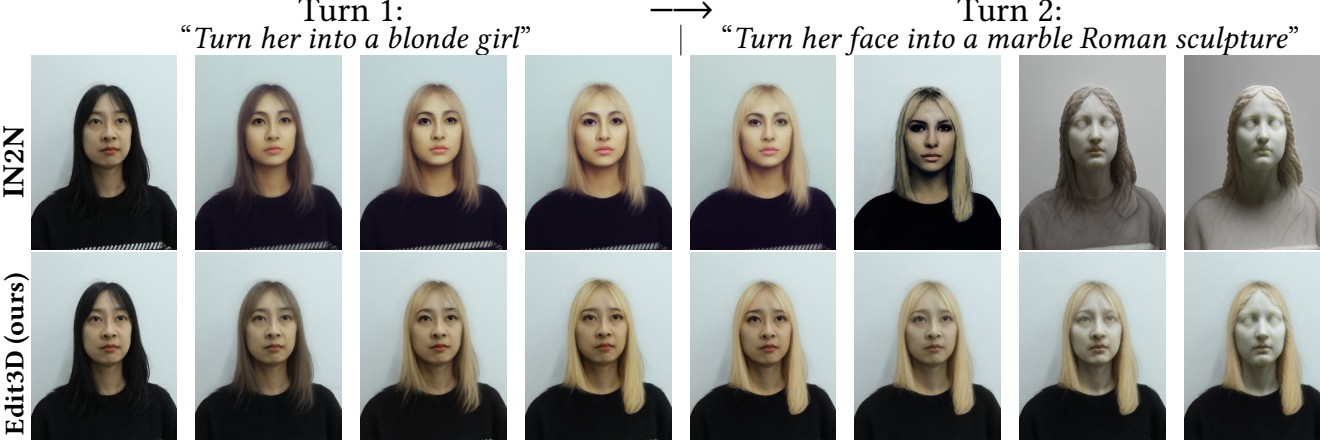

**Figure 3: Characteristic drift issue in 3D editing. The editing process is divided into two turns: the first alters the woman's hair to blonde, and the second transforms her face into a marble Roman sculpture. As the training iterations progress, the top row, marked "IN2N", displays noticeable deviation during the editing process, while the bottom row, denoted as "Edit3D", demonstrates more precise and consistent transitions through the multi-stage edits.**

images reflect the cumulative edits, our method effectively prevents the catastrophic forgetting issue in multi-turn editing. For example, as shown in Fig. 1(b), Edit3D demonstrates a seamless two-turn editing process for 3D scenes.

### 3.2 Mitigating Characteristic Drift

It is observed that IN2N exhibits a characteristic drift issue, as depicted in the top row of Fig. 3. The figure displays a 3D editing process, which is divided into two phases. In the first phase, the woman's hair color is changed to blonde, and in the second phase, her face is transformed to resemble a marble Roman sculpture. The top row, "IN2N", shows significant deviations as the training iterations progress. For instance, consider the instruction "*Turn her into a blonde girl*". As training progresses, the modified scene by IN2N adheres to this instruction. Yet, with an increase in training iterations, there's a noticeable shift in the person's identity within the edited outcomes compared to the original images. IN2N can mitigate this issue by reducing the training iterations, though this might affect the editing's quality. Take the instruction "*Turn her face into a marble Roman sculpture*" as another example. If the iterations are too few, the edit might not be pronounced enough to meet the intended effect. Conversely, too many iterations can lead to excessive interference, causing the characteristics of the whole image to change. This could lead to the whole image to resemble a statue, instead of simply modifying the face.

To address the issue of characteristic drift, one straightforward method is to limit the editing process to specific regions. This strategy helps to preserve the integrity of pixels in areas not meant to be edited. Mirzaei *et al.* [38] propose a technique to create a relevance map by spotting the differences between the predictions made by the InstructPix2Pix model when provided with and without textual instructions. They concurrently trained a relevance field while updating the NeRF to obtain relevance maps for different viewpoints.

However, Mirzaei *et al.* [38] have not made their training code publicly available. We decided to develop our own approach, utilizing attention-driven segmentation to tackle the issue of characteristic drift.

**Image Fusion with Semantic Segmentation.** Our approach uses a semantic segmentation model to identify the area for editing in the image, then blends the images before and after editing using the segmentation mask. The resulting fused image is used as the target for training the 3D representation. In particular, at the beginning of each editing session, we pre-render a set of multi-view images, $\{\hat{I}^v_{\text{rendered}} | v \in V_{\text{dataset}}\}$, utilizing the previously edited NeRF. For the current editing step, the edited image, $\hat{I}^v_{\text{edited}}$, as defined in Eq. (2), is merged with the pre-rendered image according to the formula:

$$\hat{I}^v_{\text{fused}} = \hat{I}^v_{\text{edited}} \cdot M^v + \hat{I}^v_{\text{rendered}} \cdot (1 - M^v), \tag{3}$$

where $M^v = S(\hat{I}^v_{\text{rendered}})$ is the segmentation mask produced by a semantic segmentation model $S(\cdot)$, which marks the region to be edited. To make the seams of the fused image more natural, we applied Gaussian smoothing to the edges of the mask. During the editing process, the fused image $\hat{I}^v_{\text{fused}}$ is used to iteratively update the training targets. Since $M^v$ relies solely on the image rendered prior to editing, $\hat{I}^v_{\text{rendered}}$, we can calculate it in advance at the start of each editing turn. Consequently, although our approach incorporates a semantic segmentation model, it does not significantly increase the overall time required for training.

Please note that for certain instructions such as "*Give him a cowboy hat*", the pre-rendered images $\hat{I}^v_{\text{rendered}}$ do not include the "hat" attribute. This creates a challenge for the semantic segmentation model to identify the editing area. In response, we propose to extract the mask $M^v_{\text{retained}}$ for the regions in the pre-rendered image that should stay unchanged. The mask for the editing region, $M^v$, is then obtained by $1 - M^v_{\text{retained}}$. Fig. 1(b) illustrates an example where the face and mustache are preserved while a hat is added to the top of the head. In this case, the precise location of the hat

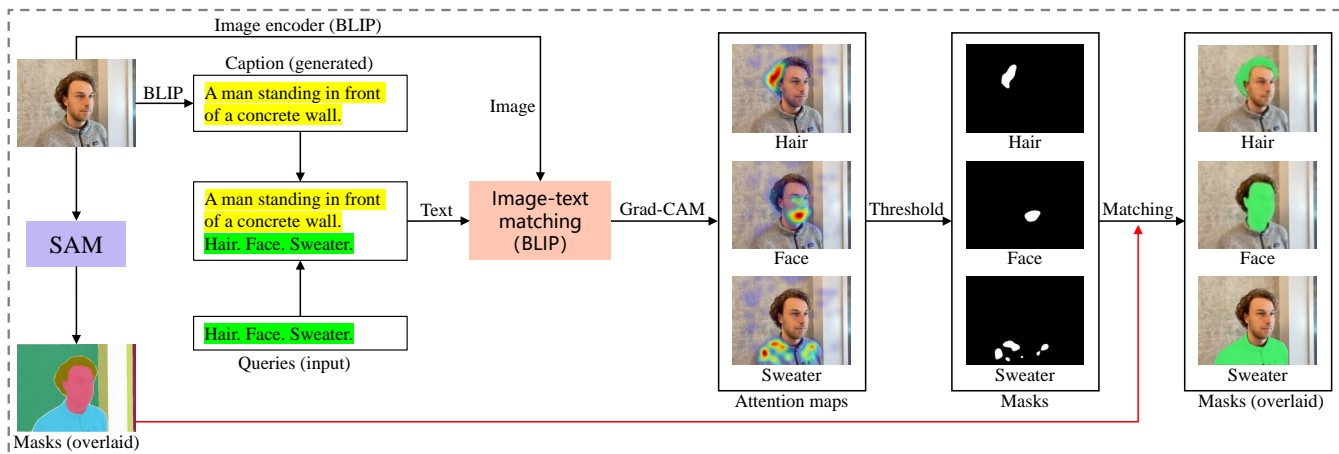

**Figure 4: Schematic of the Attention-Driven Open-Set Segmentation (ADSS) framework. ADSS employs two techniques: SAM [21] and Grad-CAM [51]. SAM generates class-agnostic masks for the entire image, while Grad-CAM extracts attention maps for text-relevant regions like hair, face, and sweater. These maps are thresholded to create masks, which are then matched with SAM's masks to achieve precise segmentation for specific parts.**

could not be determined using the segmentation model before the editing.

**ADSS: Attention-Driven Open-Set Segmentation.** Our method necessitates a semantic segmentation model that operates in an open-set manner and is adept at distinguishing various parts such as hair, face, and items of clothing like T-shirts and trousers. However, our evaluations of recent models have indicated that they do not fulfill our requirements. For instance, the model SEEM [70], as shown in Fig. 5(a), merges distinct elements such as hair, face, T-shirt, and trousers into a single segmentation mask. This issue is not unique to SEEM. We observed similar challenges with other open-set segmentation models like Grounded-SAM[1]. Additionally, our experiments with VLPart [55, 56] model anticipated to segment at the part level, exhibit shortcomings in accurately identifying faces and trousers, as illustrated in Fig. 5(b). To address these limitations, we have developed a novel model named Attention-Driven Open-Set Segmentation (ADSS), which is a part of our multi-turn editing framework. ADSS integrates two advanced techniques: SAM [21] and Grad-CAM [51].

As depicted in Fig. 4, the initial step involves deploying SAM to derive masks for the entire image, though these masks do not possess category labels. Then a caption is automatically generated for the input image using a visual-language multimodal model, BLIP [27]. This caption is subsequently concatenated with the provided text queries. The merged text and the original image are then processed through the image-text matching module of BLIP to compute attention maps for the queries via Grad-CAM, which is inspired by LAVIS [25]. We find that appending the caption before queries is crucial for generating a precise attention map for each query, as evidenced by the experiments (Sec. 4.4). For a specific query, its attention map is thresholded into a mask, which is then matched with SAM's output masks. The mask that exhibits

the largest Intersection over Union (IoU) is considered the segmentation result corresponding to the query. ADSS is an effective part-level open-set segmenter. As demonstrated in Fig. 5(c), when provided with referring texts, ADSS effectively generates masks for each respective query.

## 3.3 Enhancing Editing Quality

It has been noted that IN2N shows effective performance with images of medium resolution, such as those with dimensions up to 256 or 512 pixels. However, when it comes to high-resolution images, up to 1024 pixels, the quality of editing noticeably decreases[2]. This decline is linked to its dependency on InstructPix2Pix for image editing. InstructPix2Pix was developed using images of $256 \times 256$ resolution. Furthermore, the foundational model for InstructPix2Pix, specifically the Stable Diffusion v1.5 checkpoint, was trained using $512 \times 512$ images [48]. Therefore, while InstructPix2Pix operates efficiently at resolutions of $256 \times 256$ and $512 \times 512$, it faces challenges in adapting to higher resolutions. This limitation often leads IN2N to downsample original high-resolution images to a medium resolution during training, resulting in a loss of detailed high-frequency information in the edited images.

We propose to improve the editing effect through super-resolution. Initially, we edited low-resolution images using InstructPix2Pix and then increased the resolution of these edits with super-resolution. The resulting high-resolution images were considered the final edits and used to update the training targets iteratively. However, this approach greatly slowed down the training and was not feasible for multi-turn editing. Ultimately, we adopted a more straightforward method: once all edits were completed, we generated multi-view images from the edited NeRF and enhanced them using a super-resolution model. We further refined the edited NeRF with these high-resolution images. For superior super-resolution outcomes, we applied different models to facial and non-facial parts. We used

---

[1]https://github.com/IDEA-Research/Grounded-Segment-Anything

[2]https://github.com/ayaanzhaque/instruct-nerf2nerf/issues/14

Referring Text:

Hair | Face | T-shirt | Trousers

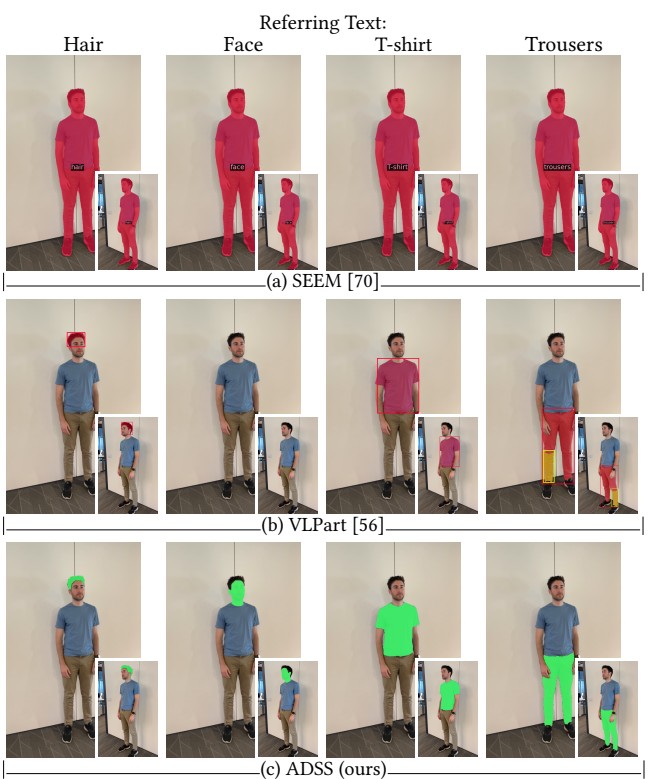

(a) SEEM [70]

(b) VLPart [56]

(c) ADSS (ours)

**Figure 5: Results of open-set segmentation. SEEM [70] colors the entire figure uniformly, unable to differentiate among hair, face, T-shirt, and trousers. VLPart [56] misses faces and shows inaccurate and inconsistent detection of trousers from various viewpoints. In contrast, ADSS demonstrates its effectiveness as a part-level segmenter.**

the GFPGAN [61] model enhanced with generative prior for facial areas, and the Real-ESRGAN [62] model for non-facial areas. This refining phase is notably efficient because it bypasses the editing process. The editing effect of the 3D scene was enhanced after refining the NeRF model.

## 4 EXPERIMENTS

### 4.1 Implementation Details

In our NeRF implementation, we selected the nerfacto model from NeRFStudio [57]. Our approach supports multiple turns of editing, with a fixed learning rate across training sessions. Initially, we train NeRF on the original scene for $30,000$ iterations, with a learning rate of $0.0003$ for camera parameters and $0.005$ for NeRF's parameters. When we move to the editing phase, we keep the learning rate for camera unchanged but lower NeRF's learning rate to $0.0025$. A typical run of $15,000$ iterations usually yields satisfactory edits. We conducted experiments with MobileSAM [65]; however, the outcomes failed to meet our expectations. Finally, we selected the SAM vit-h [21] for the segmentation model. The vision-language multi-modal model we utilize is BLIP-large [27]. We translate attention

**Table 1: Quantitative scores for multi-turn editing. Edit3D outperforms IN2N in successive edits. Please refer to Sec. 4.3 for details.**

|  | ⟨NeRF, Turn 1⟩ | ⟨Turn1, Turn 2⟩ | ⟨Turn2, Turn 3⟩ | Mean |
|---|---|---|---|---|
|  | CLIP Directional Score ↑ | | | |
| IN2N | 0.2457 | 0.1883 | 0.1607 | 0.1983 |
| Edit3D | **0.2731** | **0.2735** | **0.2088** | **0.2518** |

maps to masks using a 0.6 threshold. Our additional hyperparameters are aligned with those of IN2N, such as the adjustment of one training target every ten iterations with an edited image. Our classifier-free guidance weights for the image and text conditions are 1.5 and 7.5, respectively. Lastly, our image editing diffusion process involves 20 denoising steps.

### 4.2 Qualitative Results

We present qualitative results to evaluate the effectiveness of our proposed Edit3D, focusing on assessing the precision and consistency of edits. As shown in Fig. 6(a), we compare our method with IN2N [14] and the approach by Mirzaei *et al.* [38] (the result is from their paper due to the lack of available code). Following the editing instruction "*Give him blue hair*", the results from IN2N altered the color of the subject's eyes and clothing, which was unintended. Although Mirzaei *et al.* [38] developed a relevance field to locate the editing area, their method still affected the subject's clothing color. Our method demonstrated superior performance in adhering to the given instruction by accurately altering only the hair color while preserving the original colors of the eyes and clothing.

In Fig. 6(b) and 6(c), we showcase the editing results within the same scene by IN2N and our Edit3D, respectively. IN2N, which does not support multi-turn editing, displays a pronounced characteristic drift issue. For example, when instructed to "*Turn the red apple into a tomato*", objects irrelevant to the instruction were also transformed into tomatoes. In contrast, our method achieves precise, consecutive multi-turn editing, altering only the specific items detailed in the instructions. Fig. 6(d) presents an example of multi-turn human editing using our method. Starting from the original 3D scene, we apply a series of instructions in turn: first, we change the subject's face to resemble that of a clown, then we add a leather jacket, and finally, we shorten the trousers into shorts. Each modification is precise and impacts only the designated areas. This process demonstrates the model's capability to ensure consistency throughout the editing stages.

### 4.3 Quantitative Results

When assessing the effectiveness of multi-turn editing algorithms, the key lies in comprehensively considering the alignment between 3D editing and text instructions, as well as the coherence of the effects across multiple turns of editing. Although IN2N only provided quantitative analysis of 3D editing in two scenes, we expanded this to three different scenes and implemented three consecutive turns of editing for each scene, resulting in nine edited scenes in total. To ensure the accuracy of the assessment, we manually annotated these nine edited 3D scenes to establish the ideal text caption after

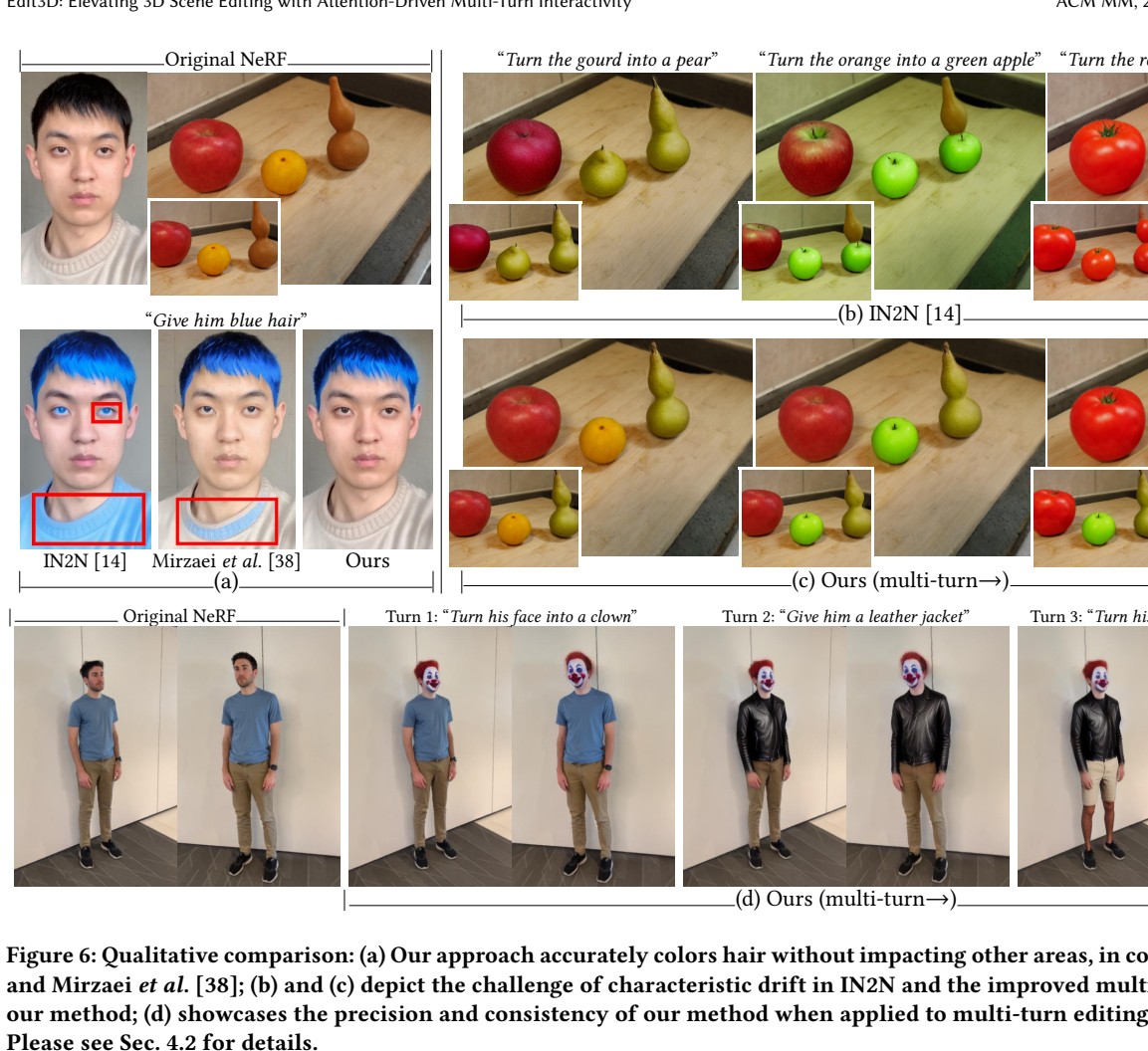

**Figure 6: Qualitative comparison: (a) Our approach accurately colors hair without impacting other areas, in contrast to IN2N [14] and Mirzaei *et al.* [38]; (b) and (c) depict the challenge of characteristic drift in IN2N and the improved multi-turn editing with our method; (d) showcases the precision and consistency of our method when applied to multi-turn editing on human figures. Please see Sec. 4.2 for details.**

**Table 2: Time efficiency was evaluated across** 10, 000 **iterations at a resolution of** 497×369 **on an NVIDIA** 3090 **GPU. See Sec. 4.3 for details.**

|  | Traing NeRF | Editing (IN2N) | Editing (Edit3D) | Fine-tuning |
|---|---|---|---|---|
| Time (minutes) | 3.48 | 44.16 | 44.62 | 4.45 |

successful editing. We employed the CLIP directional score [11] to quantify the consistency between the editing effects and the text descriptions. The score was assessed across images rendered from 100 distinct viewpoints. After completing all turns of editing, we summarized the scores obtained from each turn and calculated the average score, which served as the basis for the final assessment of the effectiveness of the editing algorithms. As depicted in Tab. 1, our approach demonstrates a clear advantage over IN2N, with a higher average CLIP directional score of 0.2518 compared to 0.1983. This indicates a more consistent alignment between textual and visual changes throughout the multi-turn editing process.

We also evaluated the runtime efficiency of our method. The results in Tab. 2 indicate that Edit3D is on par with the original IN2N. The efficiency comes from the ADSS segmentation strategy. ADSS identifies the editing areas in advance, removing the need for extra computational resources when editing. Using an NVIDIA 3090 GPU, our method can perform 10, 000 iterations in approximately 45 minutes, a duration consistent with that of IN2N. Moreover, the fine-tuning stage that employs super-resolution technology is highly efficient, completing 10, 000 iterations in under 5 minutes.

## 4.4 Ablation Studies

In this section, we examine the impact of two improvements to our editing framework: enhancement with super-resolution and segmentation with caption-prefix techniques.

*Super-Resolution Enhanced Editing.* We examine the before and after effects of fine-tuning using super-resolution. As illustrated in Fig. 7, editing without fine-tuning results in artifacts, which are marked by the red and blue boxes. With fine-tuning, the clarity of the details in the images is enhanced, showcasing the advantages of fine-tuning in the editing process.

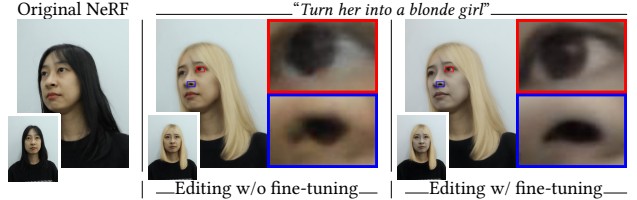

**Figure 7: Fine-tuning reduces artifacts and enhances sharpness, as evidenced by the comparisons within the red and blue boxes.**

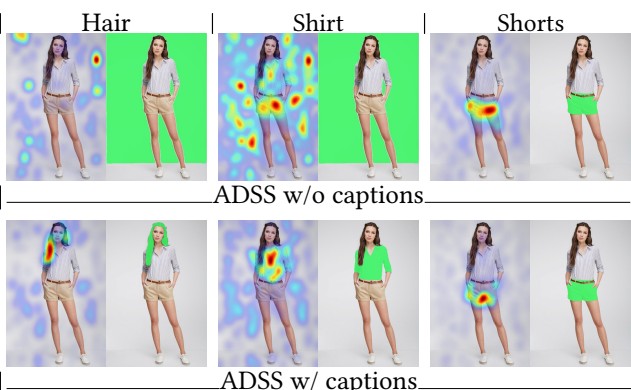

**Figure 8: Segmentation results of ADSS with and without the use of captions. Without captions (top row), the attention maps are not as precise; with captions (bottom row), there is a noticeable improvement in the attention maps, particularly in the areas of hair and shirt. The caption used for enhancement is: "A woman wearing a striped button-up shirt and khaki shorts, with long wavy brown hair".**

**Table 3: Segmentation scores for VLPart [56] and ADSS, highlighting the improved accuracy of ADSS when captions are used.**

|                    | Hair   | Shirt  | Trousers | mIoU   |
|--------------------|--------|--------|----------|--------|
| VLPart [56]        | 0.3647 | 0.9688 | 0.8753   | 0.7362 |
| ADSS w/o captions  | 0.0    | 0.0972 | 0.1938   | 0.0970 |
| ADSS w/ captions   | 0.8834 | 0.9601 | 0.9637   | **0.9357** |

*Caption-Prefix Enhanced Segmentation.* We assess the benefits of prefixing captions to text queries for segmentation. The improvements are evident in Fig. 8, where ADSS without a caption created less accurate attention maps, especially in areas like hair and shirt segments. Introducing relevant captions to the model resulted in attention maps becoming more focused, leading to more satisfactory segmentation masks. To validate this enhancement, we gathered 30 images from the DeepFashion [33] dataset and used BLIP [27] to generate image captions. Tab. 3 shows that ADSS with captions significantly outperformed its caption-less counterpart (0.9357 *vs.*

0.0970). We also tested VLPart [56], a part-level open-set segmentation model, but its accuracy was lower, particularly for hair segments, with an mIoU of only 0.3647. Additionally, we examined how the quality of captions affects segmentation performance. To do this, we added a step where the captions produced by BLIP were manually refined before being input into the ADSS model. This process made the descriptions clearer and more detailed to match the images closely. For instance, instead of generic descriptions like "a person wearing clothes", we used more detailed captions such as "a person wearing a striped shirt and khaki shorts". This refinement process led to a noticeable improvement in attention quality, showing that detailed and relevant captions are key to the success of attention-driven segmentation models.

## 5 DISCUSSION

Although our attention-driven segmentation algorithm depends on the quality of image captions, obtaining high-quality image descriptions is not a challenging task in the current technological environment. For example, before editing 3D scenes, we can generate preliminary descriptions of the 3D scenes using the open-source BLIP model, then refine the descriptions manually to improve their accuracy. Alternatively, we can directly use commercial multimodal models like GPT-4V to obtain more detailed image descriptions. Since our method relies on SAM and BLIP models, we developed a graphical UI program specifically for the image segmentation functionality to simplify the debugging process. After developing the segmentation module, we integrate it into the 3D editing framework. Despite involving several different models, the entire editing framework maintains simplicity and efficiency thanks to the modular design.

Recently, several studies have improved the effects of IN2N by defining the editing area in 3D space [5, 9, 69]. Although our method might seem less technologically advanced compared to others that operate in 3D space, the advantages of a 2D approach cannot be overlooked. Our approach simplifies the editing process, making it less challenging for users who may not have extensive 3D knowledge. As a result, it becomes more accessible to a broader audience, including those who are not professionals. Additionally, the automatic identification of editing areas in 2D space allows for faster adjustments, which is essential in cases where time is often limited.

## 6 CONCLUSION

In this paper, we introduce Edit3D, a novel attention-driven multi-turn 3D editing method. This method extends single-turn 3D editing algorithms to multiple turns and embeds a segmentation model to enhance the accuracy of the editing effects during the editing process. We also introduce a fine-tuning phase to further improve the editing results, and this phase does not significantly increase training costs. However, our method also has some limitations. For example, due to the limited image editing capabilities of Instruct-Pix2Pix [2], our method may fail when dealing with more complex editing instructions. Looking ahead, enhancing the algorithm's ability to handle increasingly complex editing instructions represents a promising research direction.

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
