# OpenReview forum: "Edit3D: Elevating 3D Scene Editing with Attention-Driven Multi-Turn Interactivity"
_acmmm.org/ACMMM/2024/Conference — MM2024 Poster_

### Official Review · Reviewer_TrSK · 2024-05-26

**Rating:** 3
**Confidence:** 4

**Summary:**

The paper presents Edit3D, a novel method for editing 3D scenes that aims to address the challenges posed by the incompatibility of new 3D representations, such as NeRF and 3D Gaussian splatting, with existing editing software. Edit3D introduces a multi-turn editing framework combined with an attention-driven open-set segmentation (ADSS) technique, allowing for more precise segmentation and minimizing interference with non-editing areas. Additionally, a fine-tuning phase is proposed to improve the overall editing quality while maintaining training efficiency. Experimental results demonstrate the effectiveness of Edit3D in adjusting 3D scenes based on textual instructions, achieving intricate combinations and enhancing the diversity of 3D editing effects.

**Strengths:**

The paper addresses a relevant and timely problem in the field of 3D scene editing, offering a potentially useful solution through the proposed Edit3D framework. The use of multi-turn interactivity and attention-driven segmentation is innovative and has the potential to improve the precision and usability of 3D editing tools.

**Limitations:**

1. The proposed method appears to be an incremental improvement over the IN2N approach. While the introduction of ADSS and multi-turn editing is interesting, the paper does not sufficiently differentiate Edit3D from existing methods in terms of fundamental innovation. The authors should provide a more detailed comparison with IN2N and highlight the unique contributions of Edit3D.
2. The experimental section lacks a thorough comparison with the latest methods, specifically GaussianEditor [5,9].  This omission makes it difficult to assess the true advancement and effectiveness of Edit3D. The authors need to include a comprehensive comparison with these state-of-the-art methods to validate their claims of superior performance.
3. In Related Work, the authors should discuss the relevant references GaussianEditor [5,9].

**Suitability:**

3

---

### Official Review · Reviewer_9oed · 2024-06-01

**Rating:** 4
**Confidence:** 4

**Summary:**

This paper proposed a 3D editing framework named Edit3D, which supports progressively precise 3d editing.

**Strengths:**

1. This paper is clear and well-written.
2. The visual experiment results are better that IN2N, especially on semantic alignment and progressive editing.
3. The ablation experiments are sufficient.

**Limitations:**

1. Lacking the discussion of related 3D editing works, e.g. DreamEditor, Progressive3D, FocalDreamer, TIP-Editor. Please cite these works and give a further discussion.
2. Edit3D claims that IN2N can not approve progressive edits. However, the main difference is Edit3D leverages the edited images as the start of next editing stage, which can be easily achieved by IN2N. Therefore, a visual ablation between Edit3D and IN2N with edited images is wanted.
3. Edit3D seems hard to add additional objects since the mask is calculated based on existing semantics.
4. The query in ADSS seems should be given by users. However, the obtainment of query and the alignment between edited regions and segment masks are desired to be automatic for convenient use.

**Suitability:**

3

---

### Official Review · Reviewer_iRsS · 2024-06-02

**Rating:** 2
**Confidence:** 4

**Summary:**

This paper proposes an attention-driven multi-round 3D editing method.
Starting from a pre-trained NeRF or other 3D representation, this method enables continuous fine-grained editing based on user input text.
The main contributions of this paper can be summarized into three parts:
1) The authors improve the IN2N optimization process to achieve a multi-round continuous editing.
2) The authors propose an Attention-Driven Open-Set Segmentation (ADSS) method, which enables precise object segmentation based on input text and attention mechanism, allowing better control over the editing area.
3) Lastly, the authors incorporated a fine-tuning super-resolution stage to enhance the final output.

**Strengths:**

* This article focuses on the need of multi-turn editing, which is important.
* The proposed method achieved satisfactory results. The supplementary video shows that the results edited by the proposed method are generally realistic and reasonable.

**Limitations:**

1. There are missing key technical details in this paper: How does the article extract the areas in the pre-rendered image that should remain unchanged?
2. Might the SR fine-tuning process introduce new inconsistencies? And this method improve the results with the pre-trained GFPGAN and Real-ESRGAN, as one of the key contributions, it seems a bit weak.
3. How is ADSS editing without captions implemented? It seems that ADSS is completely based on the input captions. And what is the editing results?
4. There are some missing ablation experiments:
* How about ablation on the improved IN2N? For example just combine the naive version of IN2N and ADSS?
* How does ADSS compare to segment models with text prompt? Although the SAM model has not yet released this feature, there is still other work available, such as SEEM: Segment Everything Everywhere All at Once (NeurIPS 2023) and Grounded-Segment-Anything.
5. How about the result on 3DGS? As the paper says it is capable of being applied to various representations.
6. What are the large and small pictures in Fig. 6 (b) and (c)？
7. The paper has a lot of redundant parts, which sometimes confuses the reader. Some descriptions can be more concise, such as sec 3.1.

**Suitability:**

3

---

### Official Review · Reviewer_3g7N · 2024-06-03

**Rating:** 4
**Confidence:** 3

**Summary:**

Edit3d proposes a new framework for multi-turn 3D editing. It has several improvements compared with InstructNerf2Nerf.

First, they propose a attention-driven openset segmentation framework for more accurate 3D local editing.

Second, they increase the resolution of the images rendered from the edited Nerf, and use them to refine the edited Nerf.

**Strengths:**

1. The writing is clear and easy to understand.
2. The improvement over InstructNerf2Nerf is obvious.
3. The attention-driven openset segmentation framework enables more fine-grained local 3D editing.

**Limitations:**

1. The comparison with InstructNerf2Nerf is unfair. As InstructNerf2Nerf is one-turn editing method, after each turn, the pre-rendered images should update accordingly. After applying this operation, It should alleviate the problem shown in Figure 1. So I suggest authors to redo the comparison experiment.

**Suitability:**

3

---

### Meta-Review · Area_Chair_39up · 2024-06-27

**Recommendation:** Accept (Poster)
**Confidence:** 5

**Metareview:**

Reviewers acknowledged the method is simple but effective. The rebuttal was successful, with two reviewers raising their scores. The overall recommendation of this paper is towards accept. After careful discussion and consideration, the ACs are pleased to inform you that your paper has been accepted. The authors should incorporate all the reviewers' suggestions into the camera-ready version. Specifically, although Reviewer iRsS increased score, they left with Borderline Reject and expressed concern w.r.t. lack of related work discussion.